# Transcriptome Profiling of the Dorsomedial Prefrontal Cortex in Suicide Victims

**DOI:** 10.3390/ijms23137067

**Published:** 2022-06-25

**Authors:** Fanni Dóra, Éva Renner, Dávid Keller, Miklós Palkovits, Árpád Dobolyi

**Affiliations:** 1Human Brain Tissue Bank, Semmelweis University, 1094 Budapest, Hungary; dora.fanni@med.semmelweis-univ.hu (F.D.); renner.eva@med.semmelweis-univ.hu (É.R.); 2Laboratory of Neuromorphology, Department of Anatomy, Histology and Embryology, Semmelweis University, 1094 Budapest, Hungary; keller.david@med.semmelweis-univ.hu; 3Human Brain Tissue Bank and Microdissection Laboratory, Semmelweis University, 1094 Budapest, Hungary; 4ELKH-ELTE Laboratory of Molecular and Systems Neurobiology, Eötvös Loránd University and Eötvös Loránd Research Network, 1117 Budapest, Hungary; 5Department of Physiology and Neurobiology, Eötvös Loránd University, 1117 Budapest, Hungary

**Keywords:** depression, prefrontal cortex, RNA sequencing, bioinformatics, gene set enrichment, protein–protein interaction network, co-expressional network analysis

## Abstract

The default mode network (DMN) plays an outstanding role in psychiatric disorders. Still, gene expressional changes in its major component, the dorsomedial prefrontal cortex (DMPFC), have not been characterized. We used RNA sequencing in postmortem DMPFC samples to investigate suicide victims compared to control subjects. 1400 genes differed using log2FC > ±1 and adjusted *p*-value < 0.05 criteria between groups. Genes associated with depressive disorder, schizophrenia and impaired cognition were strongly overexpressed in top differentially expressed genes. Protein–protein interaction and co-expressional networks coupled with gene set enrichment analysis revealed that pathways related to cytokine receptor signaling were enriched in downregulated, while glutamatergic synaptic signaling upregulated genes in suicidal individuals. A validated differentially expressed gene, which is known to be associated with mGluR5, was the N-terminal EF-hand calcium-binding protein 2 (NECAB2). In situ hybridization histochemistry and immunohistochemistry proved that NECAB2 is expressed in two different types of inhibitory neurons located in layers II-IV and VI, respectively. Our results imply extensive gene expressional alterations in the DMPFC related to suicidal behavior. Some of these genes may contribute to the altered mental state and behavior of suicide victims.

## 1. Introduction

The *dorsomedial prefrontal cortex (DMPFC)* represents one of the nodes in the default-mode network (DMN). It includes the superior frontal and the paracingulate gyrus (also called the dorsal anterior cingulate cortex) on the medial surface of the frontal cortex (*see Methods).* The DMPFC is involved in a variety of functions, combining sensory signals from the outside world transferred by the lateral prefrontal (cognitive) and the orbitofrontal (motivational) cortices, as well as viscerosensory signals through the insula and the anterior cingulate cortex [1,2]. The DMPFC prepares executive motor programs for the supplementary and presupplementary motor cortices [3,4,5]. The functions of the DMPFC include activation of working memories, decision making and planning [6,7,8,9,10]. The DMPFC is linked to social processes [11], emotional processing [12], novelty processing, dynamic contextual updating [13] and conscious threat appraisal [14,15].

The other node in the DMN is present in the medial posterior parietal cortex, including the precuneus and the posterior cingulate cortex [16,17,18]. The precuneus plays a significant role in the human brain’s mental imagination, individualization of cognitive processes, as reflections of signals from the external world, introspection, self-awareness, creativity and intelligence [19,20,21,22,23]. The precuneus directly projects to the supplementary and the presupplementary motor cortices [24,25]. In addition, the precuneus also participates in the control of the motor executive program through connections with the DMPFC [26]. 

One of the outputs of the DMN is connected to the cortico–striato (including nucleus accumbens/ventral striatum)–thalamocortical loop (circuit) that connects the DMPFC and the precuneus with the reward-association system [3,17,27]. Dysfunction in the reward-association system results in major depression [17,28,29].

Abnormal communication within the DMN has been reported in depression using resting state functional connectivity (RSFC) analysis [30]. Meta-analysis revealed that clinical response to several different treatments, including repetitive transcranial magnetic stimulation (TMS), pharmacotherapy, cognitive behavioral therapy, electroconvulsive therapy and transcutaneous vagal nerve stimulation, could be predicted by baseline DMN connectivity in patients with depression [31]. In the DMPFC, depressed patients demonstrated increased functional connectivity with sensorimotor, visual and salience network regions [32]. Furthermore, a major symptom of depression, anhedonia, was associated with increased RSFC between seed regions of bilateral nucleus accumbens and areas of right DMPFC [33]. Another recent line of research, which suggest that the DMPFC may be involved in depression, is that repetitive TMS of the DMPFC was found as a novel intervention for treatment-refractory depression [34,35]. 

Although depression is the most common psychiatric disorder in people who die by suicide [36,37,38], there may also be differences in the underlying mechanisms. A study found evidence of different patterns of connectivity strength within the DMN of depressed–suicidal and depressed–nonsuicidal adult participants [39]. The involvement of the DMN in suicide is predicated on their key role in self-referential thinking, which may be a consequence of altered integrity within the DMN. On the basis of subregions previously reported to show structural and functional alterations within the DMN, functional activation studies applying cognitive or motor tasks generally showed altered involvement of the DMPFC linked with suicidal thoughts among depressed individuals [40,41,42].

Genetic variants of serotonergic, dopaminergic and adrenergic genes are the most extensively studied genes in relation to both depression and suicidal behavior; however, to date, only few candidate gene variants have been reliably associated with suicidality [43]. A recent meta-analysis identified novel depression-associated variants involved in the development and maintenance of cognitive processes [44]; moreover, a systematic review revealed that the most crucial candidate genes for major depressive disorder (MDD) are involved in glutamate neurotransmission, regulation of calcium channel activity and apoptosis [45]. A cell-type-specific methylome study pointed to the role of the innate immune responses via p75NTR/neurotrophic growth factor and Toll-like receptor signaling in MDD [46]. Neuronal plasticity, and other pathological alterations associated with depression and suicidal behavior, may involve gene expressional alterations [47]. Initially, individual systems such as GABAergic function [48], neurotrophic factors [49] and glial cells [50] were examined. More recently, high-throughput methods were applied to establish a wider variety of genes significantly altered in suicide victims. Microarray studies identified a number of altered genes, regulating, e.g., glial, endothelial and mitochondrial activities [51] some of them showing sexual dimorphism [52] and comorbidity with substance use disorder [53,54]. More recently, sequencing approaches identified additional new susceptibility genes and pathways in depression using animal models of chronic stress [55] and also human studies related to depression [56] and suicide [57]. Although these gene expressional studies provided valuable data on the neuronal mechanisms of suicidal behavior, they were all performed in the dorsolateral prefrontal cortex (DLPFC). Given the above-described roles of the DMPFC, we hypothesized that gene expressional changes associated with suicidal behavior take place in the DMPFC as well. We obtained DMPFC samples of suicide victims with relatively short postmortem delay whose investigation provided useful insight into the underlying reasons of suicidal behavior. 

## 2. Results

### 2.1. Transcriptome Sequencing in the DMPFC of the Suicide Brains

The mean age, sex ratios and *post mortem* interval (PMI) did not show detectable differences between the suicide and control groups (eight individuals/groups) (see Methods, Appendix A). On average, 62.9 ± 3.6 Mb total raw reads per sample were generated. The reads demonstrating inadequate-quality, adaptor-containing or unknown sequences were not included in downstream analysis. The average clean reads ratio after quality control (QC) was 91.87 ± 0.46%, and the average mapping ratio with the reference genome was 91.28 ± 0.79%. In parallel, Q20 and Q30 (the percentage of bases number which is higher than 20 and 30 in the total number of bases, respectively) of the clean data were calculated (Appendix A). 

The average ratio of clean reads Q20 was 97.46 ± 0.079%, and the average ratio of clean reads Q30 was 89.80 ± 0.19%. These analyses indicated the high quality of library construction and sequencing data of postmortem human samples. The average mapping ratio with genes was 72.22 ± 0.96%. These high alignment percentages suggest the suitability of the sequenced data for downstream analysis. In total, 29 791 genes were identified, out of which 28 088 were known genes. About 67% of the fragments were attributed to protein-coding genes. The remaining fragments were attributed to other RNA classes, including long intergenic noncoding RNAs, pseudogenes, etc. The amount of total raw reads, clean reads and their percentages are shown in Appendix A.

### 2.2. Comparison of Suicide and Control Samples Based on Differentially Expressed Genes (DEGs)

After counting the reads aligning to each gene, we used DESeq2 to analyze gene expression in suicide victims compared to matched controls. We detected expression of 19,692 protein-coding genes in the DMPFC, out of which 1400 were differentially expressed between the two groups, 1262 genes were downregulated and 138 genes were upregulated at the adjusted *p*-value < 0.05 and log2 FC > ±1 (Appendix A). All samples were compared to one another using Euclidean distance. The correlation matrix based on all gene expression data indicated that samples mostly cluster by individual and diagnostic group with three control samples showing some similarity to the suicide samples (Figure 1A). We compared the age, sex and PMI between the three controls clustered with the suicide victims and those who were not clustered using a *t*-test. We found no significant differences between the two groups of control individuals for any of these covariates (age: *p* = 0.51; PMI: *p* = 0.39; sex ratio: *p* = 0.67). Therefore, we do not assume any systematic confounding variables that can distort the results. Hierarchical clustering of suicide and control samples using 1400 differentially expressed genes revealed that these genes also distinguished the two groups (Figure 1B). 

Among the DEGs, 1262 genes were downregulated and 138 were upregulated in suicide victims, as shown in the volcano plot (Figure 2).

### 2.3. Functional Annotation and Classification of the DEGs

To understand how the DEGs from the DMPFC relate to suicidal behavior, Gene Ontology (GO) classification was performed. Figure 3A shows the top 10 down- and upregulated DEGs. The top three downregulated genes were CSF3, IL1R2 and SOCS3. They all play a critical role in inflammation. The top three upregulated genes were P2RX2, ATP4A and BX276092.9. P2RX2 and ATP4A are found in the plasma membrane and both of them have a role in ATP binding and ion transport. BX276092.9 is a barely characterized protein. Additional GO functional enrichment analysis identified significant differences in more than 500 functional pathways of downregulated genes and 37 functional pathways of upregulated genes at adjusted *p*-value < 0.05 (Appendix A). A comparison with the Allen Human Brain Atlas (AHBA) expression database showed that some of the altered genes are expressed in specific cell types. For example, one of the top downregulated genes, the MT-ND5, is produced in FEZF2 and RORB expressing excitatory and SST+ inhibitory neurons. The LORICRIN, which is one of the top upregulated genes, is expressed in THEMIS expressing excitatory neurons (Appendix A). 

The top three enriched GO terms within the biological process, molecular function, and cellular component categories were visualized (Figure 3B,C). A number of pathways related to cell surface receptor signaling (GO:0007166) and growth factor binding (GO:0019838) were enriched in downregulated genes in suicidal individuals. In contrast, pathways pertaining to synaptic signaling (GO:0099536) and sodium channel activity (GO:0005272) were enriched in upregulated genes of suicidal individuals (Figure 3B,C). To further identify the pathways involved in suicidal behavior, pathway analysis of DEGs was performed using pathway classifications from the Kyoto Encyclopedia of Genes and Genomes (KEGG) and Reactome resources (Statistics of Pathway Enrichment). The top three enriched KEGG and Reactome pathways are shown in Figure 3D. A total of 23 significantly enriched downregulated KEGG pathways and 25 significantly enriched downregulated Reactome pathways were found in downregulated genes, while there was one significantly enriched upregulated KEGG pathway and two upregulated Reactome pathways at adjusted *p*-value < 0.05. The complete list of significantly enriched pathways can be found in Appendix A.

### 2.4. Protein–Protein Interaction Analysis of DEGs and Identification of Key Genes

Computational methods analyzing protein protein interaction (PPI) networks are useful for understanding the biological meaning of gene expressional alterations. Therefore, we used the STRING online database (http://string-db.org, (accessed on 11 November 2021)) to construct the PPI networks of down- and upregulated DEGs, which were then transferred into the Cytoscape software. Analysis of top DEGs was achieved using the degree method in cytoHubba where the top 10 down- and upregulated DEGs were identified (Figure 4A,C). Enrichment analysis of the top 10 downregulated DEGs s (Figure 4A) through stringApp revealed that these genes are mainly associated with astrocyte differentiation (GO:0048708), regulation of MAPK cascade (GO:0043408) and cell surface receptor signaling pathway (GO:0007166) (Figure 4B); meanwhile, the top 10 upregulated DEGs (Figure 4C) were associated with regulation of membrane potential (GO:0042391), nervous system process (GO:0050877) and cell–cell signaling (GO:0007267) (Figure 4D). The complete list of genes and their associated pathways can be found in Appendix A. Using the AHBA database for comparison, we found that most of the top downregulated genes are abundantly expressed in astrocytes. In turn, the top 10 upregulated genes are expressed in excitatory and inhibitory neurons and can be also found in oligodendrocytes (Appendix A).

### 2.5. Validation of RNA-Seq Data

To verify changes in gene expression associated with suicidal behavior, we performed qRT-PCR. We have chosen fourteen functionally relevant genes (six up- and eight downregulated) from the DEGs, which have been potentially implicated in depressive behavior based on the literature: GRIK1, GRIK2, GRM2, NRGN, SYT5 and NECAB2 as upregulated genes and AQP1, ITPKB, ITGB4, SLCO2B1, GJA1, PRKCH, GLUL and S100B as downregulated genes. As shown in Appendix A, results of quantitative PCR analysis of the selected genes confirmed the RNA-seq results. Thus, we verified the increased expression of GRIK1, GRIK2 and GRM2 involved in glutamate signaling, as well as NECAB2, while the reduced expression of astrocyte-related genes, such as S100B, AQP1, GJA1, GLUL, ITPKB and ITGB4 in suicide victims (Table 1, Appendix A).

### 2.6. Depression-Focused Gene Set Enrichment

The top 10 down- and upregulated DEGs based on fold change as well as the PPI and co-expression network analyses were investigated in the DisGeNET database to search for significant enrichments associated with depression and other mental disorders using the disease classification terms “Mental Disorders” and “Behavior and Behavior Mechanisms”. We found 95 enriched categories associated with more than one gene. We found 24 genes that have been previously associated with depression, including CSF3, IL1R2, IL6, SERPINA3, MT-ND5, MT-ND6, P2RX2, ATP4, NOTCH1, EGFR, TLR4, STAT3, ERBB2, PROC, CARTPT, GRIK2, CACNA1G, GABRD, BCL2, CHDH, NRGN, SERPINF1, CCK and CRHR1. Genes associated with depressive disorders, schizophrenia and impaired cognition were strongly enriched in the DMPFC. Furthermore, we found that the upregulated genes CCK, GRIK2 and CRHR1 are affected in suicide (Appendix A). Although the database does not contain information specific to DMPFC, our data suggest that these genes may be associated with suicide in this brain region, too.

### 2.7. Co-Expression Network Analysis and Hub Gene Screening in the DMPFC of Suicidal Individuals

We built two independent gene co-expression networks for down- and upregulated genes to identify gene co-expression communities (Appendix A). Analysis was conducted using highly correlated gene pairs with Pearson correlation > 0.9 and Padj < 0.01. In the networks, the DEGs were represented by nodes and pairwise co-expression relationships between genes by edges. The co-expression network of downregulated DEGs revealed the top 10 highest ranked hub genes as SORBS3, RHOC, S100A16, SZRD1, TRIP6, AHNAK, GRIN2C, CHDH, MAPKAPK2 and FTL with higher degree and betweenness centrality than others, indicating a more critical role played by them in the network. The network was further subdivided into 21 functional clusters composed of a total of 710 downregulated DEGs (nodes) (Appendix A). 

There was a significant difference in the biological processes as different modules were enriched (Figure 5A). For example, the biggest module Cluster 1 (1) was significantly enriched using biological terms of the GO classification, such as “gliogenesis” (GO:0042063) and “response to cholesterol” (GO:0070723). The second biggest module, Cluster 2 (2) was significantly enriched using biological terms, such as “regulation of epithelial cell differentiation” (GO:0030856) and “type I interferon signaling pathway” (GO:0060337) (Figure 5A). The co-expression network of upregulated DEGs revealed the top 10 most ranked hub genes as CALY, SOHLH1, CPNE9, NRGN, MAST1, GRIK1, SYT5, TRPM2, BX276092.9 and MYO15A. The network was further subdivided into six functional clusters of a total of 97 upregulated DEGs (Appendix A), where Cluster 1 (1) was significantly enriched using biological terms “neurotransmitter receptor internalization” (GO:0099590) and “biotin metabolic process” (GO:0006768). Cluster 2 (2) was significantly enriched using biological terms, such as “sodium channel activity” (GO:0005272) and “neuropeptide hormone activity” (GO:0005184) (Figure 5B). The hub proteins of clusters were specifically identified as most ranked, based on their degree and betweenness centrality (Appendix A). For instance, the hub protein of downregulated co-expression network in Cluster 1 is the SORBS3 encoding vinexin protein, which is implicated in promoting the upregulation of actin stress fiber formation. An additional hub protein of the downregulated network in Cluster 2 is RHOC encoding the Rho-related GTP-binding protein, which is known to regulate the signal transduction pathway linking plasma membrane receptors to actin stress fibers. The hub protein of upregulated co-expression network in Cluster 1 is CALY encoding the neuron-specific vesicular protein calcyon, which interacts with the clathrin light chain A and stimulates clathrin-mediated endocytosis. Another hub protein from the upregulated network in Cluster 2 is SOHLH1 encoding the spermatogenesis- and oogenesis-specific basic helix-loop-helix-containing protein 1, which was implicated in transcriptional regulation of both male and female germline differentiation. The entire co-expression networks and the annotation information of the top three clusters are available in Appendix A.

Using the CytoHubba plugin in Cytoscape, we determined the top 10 hub genes of down- and upregulated co-expression networks according to their highest Maximal Clique Centrality (MCC) scores (Figure 6A,B). We constructed a protein–protein interaction network (PPI) for all the top 10 down- and upregulated hub genes using the STRING database, out of which two of the downregulated and four of the upregulated genes were connected to each other (Figure 6C). In GO and Reactome pathway enrichment analyses of hub genes, we identified enriched gene sets related to developmental processes, such as actin filament organization, neuron maturation or anterograde axonal transport (Figure 6D). We found similar results when analyzing cell type expressing features of the top hub genes by using the AHBA database, where most of the downregulated hub genes are expressed in astrocytes, while the upregulated ones can be found in excitatory and inhibitory neurons and oligodendrocytes (Appendix A). The top 10 hub genes from the co-expression networks and their functional annotation are listed in Appendix A.

### 2.8. Distribution of NECAB2 in the DMPFC

As a functionally intriguing validated DEG, NECAB2 was selected for analysis of its distribution within the DMPFC. In situ hybridization histochemistry was applied in a control and a suicidal individual to localize the expression of NECAB2 mRNA. The hybridization signal was abundant in layers II–VI (Figure 7A,B). NECAB2 immunolabeled cells showed the same distribution (Figure 7C). Immunolabeling also revealed that NECAB2 is present in two different interneuron subtypes. Most of the NECAB2-positive interneurons had small size cell bodies, while some of them demonstrated large soma (Figure 7(C3)). The larger interneurons were located in deep layers; meanwhile, the smaller ones were distributed in the upper layers, predominantly in layer II (Figure 7(C1,C3)). Immunolabeling also revealed that NECAB2 is present not only in the somato-dendritic compartments, but also in axon terminals in neurons (Figure 7(C2,C3)). To determine whether NECAB2 protein is also present in glial cells, we performed combined immunolabeling for Iba1 (a microglia marker) and S100B (an astrocyte marker) with NECAB2. The lack of colocalization indicates that NECAB2 is not expressed in glial cells (Figure 7D–F).

## 3. Discussion

The underlying pathophysiological basis of suicidal behavior remains basically obscure. Neuroimaging studies revealed that committing suicide, similar to MDD, may be a neural network-level disturbance [58,59]. Neuroimaging studies revealed abnormal resting-state functional connectivity (RSFC) between distributed brain areas in MDD. The connection between the medial prefrontal cortex and the medial posterior parietal cortex has been named the resting-state network (RSN) since they have temporal correlation in the absence of a variety of tasks in resting condition [60,61,62,63]. The RSN includes distinct networks, such as the *default mode*, *salience* and *attention networks* [3,9,16,64,65,66,67]. The default mode network (DMN) has been highlighted in neuroimaging studies, since its discovery has attracted increased interest due to its implications for MDD and suicide-related behavior. The DMN is involved through convergent findings of increased resting-state functional connectivity between two core DMN regions, namely the DMPFC and precuneus [68].

Even though the DMPFC is a major component of the DMN, no attention has been assigned to examine transcriptome changes in this region in suicide victims or depressed patients [51,53,57]. We present here the results of RNA-seq analyses in the DMPFC from sixteen subjects, eight with fatal suicide action and eight controls. The number of samples in both groups are sufficiently high to conclude significant results and draw proper conclusions [69]. We identified more than 1000 DEGs. The number of DEGs is in line with previous studies investigating suicide victims [53] and schizophrenic individuals [70]. In our study, a high number of genes decreased their expression level. Since our RNA-seq data fit the DESeq2 model, technical issues are not likely. Therefore, we conclude that suicidal behavior is accompanied by a generally reduced gene expression in the DMPFC.

### 3.1. Conclusions Based on Individual DEGs

Among the DEGs the top 10 downregulated genes were involved in immune response, neurodegeneration, mitochondrial electron transport and cell adhesion. The role of immune dysfunction in suicide victims has been reported in the prefrontal cortex [51] and the insula [71]. It has also been proposed that neurodegeneration and impaired structural neuroplasticity [71,72], as well as mitochondrial dysfunction [53,73], are associated with suicide completion. The top 10 upregulated genes were involved in ATP signaling, protein and vesicle-mediated transport, and were components of membrane and cytoskeleton and regulating protein secretion. Changes in protein and vesicle-mediated transport in suicide have also been observed previously [51,71,74].

In addition to the DEGs with the highest fold changes, we also focused on the individually validated DEGs. For qRT-PCR validation, we selected 14 genes whose alterations were reported in human depression and suicide or rodent depression-like behavior [52,54,75,76,77,78,79,80]. For ITPKB, GJA1 and GLUL genes, the previously reported alteration in the DLPFC [54] was demonstrated to also occur in the DMPFC. For AQP1 and ITGB4, studied only in rodent models of depression [75,78], the present study was the first to implicate them in human suicidal behavior.

Several genes connected to astrocyte function, such as ATP1A2, ALDH1L1, GFAP, S100B, GJA1 and AQP1, showed decreased expression levels in suicide victims. These observations suggest that genes promoting astrocyte functions may be involved in the pathophysiology of depression and suicidal behavior, as shown previously for depression [81,82,83].

### 3.2. Conclusion Based on the Distribution of NECAB2

NECAB2 is one of the validated genes whose level increased in suicide victims. It was shown to bind specifically to the type 5 metabotropic glutamate receptor (mGluR5) to modulate its function [84]. Our findings, consistent with a previous study [85], provided evidence that the glutamatergic signaling system is involved in the pathogenesis of suicidal behavior. Therefore, it is likely that NECAB2 exerts its role on suicidal behavior by increasing the activity of mGluR5. The AHBA suggests that NECAB2 is expressed in interneurons located in layers II-VI in the human cerebral cortex. The majority of NECAB2 expressing cells are GABAergic interneurons belonging to the vasoactive intestinal peptide (VIP), lysosome-associated membrane glycoprotein 5 (LAMP5), paired box 6 (PAX6) and somatostatin (SST)-expressing interneurons (Appendix A). Our in situ hybridization and immunohistochemistry study confirmed that NECAB2 is not expressed in glial cells. We found that NECAB2 was presented in the cell body of two morphologically different interneuron subtypes in the DMPFC. The interneurons with larger bodies were located in deep layers, while the smaller, more abundant cell types were distributed in upper layers, mainly in layer II. Based on the distributional and morphological data, the larger cell type may correspond to SST+ interneurons while the smaller one corresponds to VIP+ interneurons [86,87]. The presence of NECAB2 in somatodendritic compartments, as well as in axon terminals, in DMPFC interneurons is consistent with previous data on the hippocampus [88]. Based on these data, it is likely that at least one of the cell types expressing NECAB2 is implicated in suicidal behavior, possibly due to the regulation of mGluR5.

### 3.3. Functional Implications Based on Pathway Analyses

Our integrated analyses using GO functional annotation, as well as KEGG and Reactome pathways revealed a number of pathways altered in suicide victims. The downregulated DEGs were significantly enriched in the cell surface receptor signaling pathway of biological process and growth factor binding of molecular function, in PI3K-Akt signaling pathway, in TNF signaling pathway and in cytokine–cytokine receptor interaction. Alterations in inflammatory responses have been related to mental illnesses [89,90,91]. However, our study is the first to suggest that suicidal behavior is linked to reduced inflammatory ability in the DMPFC, suggesting a reduced microglial function in suicidal behavior.

Upregulated DEGs were enriched in axons and synapses as cell components, in synaptic function as biological process and in sodium channel activity as molecular function. Pathways found among upregulated DEGs were neuroactive ligand–receptor interactions and activation of Na-permeable kainate receptor pathways, which all suggest increased glutamatergic, specifically kainate, signaling. This change has been implicated in mood disorders [92,93], addictive disorders [94] and in alcohol dependence [95]. Further evidence from postmortem studies revealed higher expression levels of the majority of glutamatergic genes in the DLPFC associated with MDD and suicide [74,96], and now we provide evidence that it also holds for the DMPFC. Target genes of high interest included GRIK2, which likely plays a role in emerging suicidal thoughts after antidepressant treatment [97]. In a subsequent study, polymorphisms in GRIK2 gene were further associated with suicidal ideation in MDD patients following antidepressant treatment [98]. A rodent study showed that GRIK2-deficient mice were more impulsive and aggressive, suggesting that it has a unique role in controlling the behavioral symptoms of mania [99]. In particular, the outcome of completed suicide has been associated with increased expression of GRIK2 [73]. The present data further suggest that elevated kainate receptor signaling in the DMPFC can predispose the development of committing suicide, and hence can be a potential biomarker.

Functional annotation of down- and upregulated DEGs revealed that there are gene sets overlapping common pathways associated with depression. Searching for genes, which share common functions with depression-related pathways, we found that both down- and upregulated genes were present, which participate in neurotransmission and cell metabolism processes taking place in the DMPFC (Table 2).

### 3.4. Functional Cluster Analysis of Gene Expressions in the DMPFC

Co-expression analysis on the DEGs was performed to identify the key modules of highly co-expressed genes. The largest cluster in a downregulated gene network was associated with gliogenesis, while another cluster was associated with type-I interferon signaling pathway, suggesting reduced glial function and the role of interferon gamma in reduced inflammatory response ability. In turn, modules in upregulated gene networks were associated with neurotransmitter receptor internalization, sodium channel activity and peptide neuromodulator function. Alteration in the neurotransmitter receptor internalization pathway is consistent with the assumption that dysfunction of neurotransmitter receptors may contribute to the pathophysiology of MDD [93,100,101].

Intramodular hubs have an important role in the enriched pathways of the modules in co-expression networks. The major hub gene of Cluster 1 in the downregulated gene network was SORBS3 (vinexin). This protein modulates the actin cytoskeleton and negatively regulates autophagy. Its expression increases with age in mouse and human brain tissue, contributing to autophagic decline in mammalian brain aging [102,103]. Moreover, a meta-analysis provided evidence that SORB3 has a strong relationship with MDD [104].

The hub gene from upregulated gene network, calcyon-neuron-specific vesicular protein (CALY) regulates dopamine-related signaling [105] and dopamine D1 receptor internalization of pyramidal cells in the prefrontal cortex and dorsal striatum [106]. Other studies reported that CALY has trafficking functions primarily involved with neural development and synaptic plasticity [107,108]. These results suggest that intramodular hub genes could function as potential biomarkers for future therapeutic interventions.

The top 10 suicide-related down- and upregulated hub genes were screened from the co-expression networks according to the MCC scores, including the downregulated genes MSI2, SNTA1, S100A16, RHOC and SORBS3, and the upregulated genes ACTL6B, CCK, CALY, CPNE9 and SYT5. One of the top downregulated hub genes, the Musashi-2 (MSI2) RNA-binding protein, was found to be expressed in neural progenitor cells [109] and involved in the regulation of cell cycle and development [110]. Another downregulated hub gene, the alpha-1-syntrophin (SNTA1), plays an important role in the organization of the localization of a variety of membrane proteins and establishes a link between the extracellular matrix and receptors [111,112]. Between the upregulated hub genes, the actin-like protein 6B (ACTL6B) belongs to the neuron-specific chromatin remodeling complex, and thus plays a crucial role in neural development and dendritic outgrowth [113]. A defect in neuronal dendritic branching was observed in ACTL6B knockout iPSC-derived neuronal cells [114]. Another upregulated hub gene is the cholecystokinin (CCK) produced in the gastrointestinal tract. It acts as a pancreatic enzyme to regulate digestion; however, it has been also identified as a neuropeptide abundantly expressed in vertebrate brains [115]. The functional and pathway enrichment analysis found integral biological processes enriched in terms related to cell cycle regulation, cell motility and signal transduction.

### 3.5. Functions Supported by Known PPIs of DEGs

Analysis of the top 10 most ranked downregulated genes (PECAM1, ERBB2, ITGB1, EGFR, STAT3, NOTCH1, CD44, IL6, TLR4 and FN1) from the PPI network revealed that these genes were strongly enriched in cell-surface receptor signaling pathway, astrocyte differentiation and regulation of MAPK cascade, suggesting that these pathways might be implicated in suicide behavior. As previously defined, two of the top most ranked genes, the adhesive stress-response protein PECAM1 and the proto-oncogene ERBB2 were reported as decreasing in the DLPFC in suicide victims [54]. Analysis of top 10 most ranked upregulated genes (CCK, GABRD, CRHR1, CALY, COL11A2, COL24A1, SCN3B, GRIK1, GRIK2, CACNA1G) revealed that these genes were associated with regulation of membrane potential, nervous system processes and cell–cell signaling. A pilot study examining the GABAergic system in suicide victims found GABRD variants in the prefrontal cortex of patients associated with depressive disorder [116]. Recently, a strong association between CRHR1 polymorphism and suicide attempts was shown [117]. Löfberg et al. [118] found that MDD patients associated with the number of previous suicide attempts had higher CCK levels. A more recent study showed that CCK levels were higher among first suicide attempters after the first 12 h following the attempt [119]. Likewise, a study investigated postmortem brain tissues of completed suicides observed higher number of CCK receptors in the frontal cortex compared to healthy controls [120].

A multi-locus GWAS study demonstrated that several allelic variations in glutamatergic synaptic signaling, such as GRIK1, GRIK2, GRIK3, GRIN1, GRIN2A, GRIN2C or GRM7, were associated with MDD [77]. A subsequent systematic review revealed that several crucial candidate genes for MDD are involved in glutamate neurotransmission, [45]. Moreover, several studies found decreases in mRNA expression of NMDA and AMPA receptor subunits associated with schizophrenia in the frontal cortex [121,122,123]. The involvement of glutamate in psychiatric and medical conditions has been intensively examined; however, earlier studies mostly focused on the biology and pathophysiology of ionotropic glutamate receptors. In turn, metabotropic receptors (mGluR) can also modify neuronal activity through G-protein coupled signaling. Indeed, strong interactions have been reported between mGluR5 and NMDA receptors, suggesting that mGluR5 might be implicated in mediating neural plasticity as well as learning and memory processes [124]. Two proteins from upregulated DEGs, the CACNG8 and CALY, are considered neurotransmitter receptor regulatory proteins. CALY regulates dopamine D1 receptor internalization in the prefrontal cortex and dorsal striatum [106], and CACNG8 (also known as TARP-γ8) negatively modulates AMPA receptor signaling [125].

### 3.6. Genes Associated with Depression and Comorbidities

Among the top DEGs, 10 genes have been previously associated with depression, including CSF3, IL1R2, IL6, SERPINA3, MT-ND5, MT-ND6, P2RX2, ATP4, PROC and CARTPT, based on the DisGeNET database [126]. Studies of genetic polymorphisms in psychiatric disorders have shown that CARTPT gene mutation exhibits increased anxiety and depression [127]. Furthermore, the peptide encoded by CARTPT is a candidate biomarker for MDD because of its effects on mood regulation [128,129]. Additionally, we identified 12 genes from the most ranked down- and upregulated DEGs associated with depression, such as GABRD, IL6, NOTCH1, EGFR, ERBB2, TLR4, STAT3, GRIK1, GRIK2, CACNA1G, CCK and CRHR1 and six genes from the hub gene list of the co-expression analysis, including BCL2, CHDH, GABRD, NRGN, SERPINF1 and CCK.

## 4. Materials and Methods

### 4.1. Human Brain Tissue Samples

Human brain samples were collected in accordance with the Ethical Rules for Using Human Tissues for Medical Research in Hungary (HM 34/1999) and the Code of Ethics of the World Medical Association (Declaration of Helsinki). Tissue samples were taken during brain autopsy at the Department of Forensic Medicine of Semmelweis University in the framework of the Human Brain Tissue Bank (HBTB), Budapest. The activity of the HBTB has been authorized by the Committee of Science and Research Ethic of the Ministry of Health Hungary (ETT TUKEB: 189/KO/02.6008/2002/ETT) and the Semmelweis University Regional Committee of Science and Research Ethics (No. 32/1992/TUKEB). The study reported in the manuscript was performed according to a protocol approved by the Committee of Science and Research Ethics, Semmelweis University (TUKEB 189/2015). The medical history of the subjects was obtained from clinical records, interviews with family members and relatives, as well as pathological and neuropathological reports. All personal data are stored in strict ethical control, and samples were coded before the analyses of tissue.

### 4.2. Tissue Preparation

Postmortem human brain tissue samples from the dorsomedial prefrontal cortex (DMPFC) (Brodmann area 9) were acquired from the Human Brain Tissue Bank (Semmelweis University, Budapest, Hungary). There is considerable variability in how the topographical extension of the DMPFC is defined. The medial prefrontal cortex (MPFC), as its name indicates, occupies the medial surface of the frontal lobe over the anterior cingulate gyrus. The MPFC is divided into dorsal and ventral prefrontal cortex with a theoretical horizontal line through the most rostral (genual) point of the corpus callosum. (These two parts are frequently called anterior and posterior prefrontal cortex.) The posterior border of the DMPFC is defined by the precentral sulcus, which separates it from the premotor cortex. The border of the dorsomedial and dorsolateral prefrontal cortex in coronal sections can be defined by the deep superior frontal sulcus (Figure 8). 

The white matter also serves as an excellent landmark: the superior longitudinal cortical pathway, as a well visible white matter bundles, are divided into a medial and a lateral portion in the dorsal prefrontal cortex. The cortical samples were collected from 16 subjects: 8 control subjects with no history of psychiatric or neurological diseases (2 females and 6 males, mean age of 65.4 ± 5.6) and 8 suicide victims (3 females and 5 males, mean age of 53.6 ± 4.8). The age of the subjects ranged from 31 to 89 years (Table 3). The selected control subjects were not diagnosed with any psychiatric disorder, while the suicide victims were without evidence of acute or chronic depression. Brains were removed from the skull with a postmortem delay of 1 to 10 h, frozen rapidly on dry ice and stored at −80 °C until microdissection. Serial coronal sections were cut from the frontal lobe. Sections between 50.0 and 30.0 mm from the origin of AC-PC coordinates (see [130]) (practically at the first appearance of the genu corporis callosi or the first appearance of the anterior pole of the lateral ventricle) were punched out. Special microdissection needles with 8 and 15 mm inside diameters were used [131,132]. Tissue pellets (1–3 per brain) include samples from the superior frontal gyrus (Brodmann area 9) (both gray and white portions within the gyrus) and the paracingulate cortex (also called the dorsal part of the pre- and supragenual portions of the anterior cingulate cortex, Brodmann area 32), the upper bank and the deep portion of the gyrus (Figure 8). In samples, the superior frontal gyrus/paracingulate gyrus ratio was about 80–85%, respectively. The microdissected tissue pellets were collected in 1.5 mL Eppendorf tubes and kept until use at −80 °C. Throughout the microdissection procedure (slicing, micropunch, storage) the tissue samples were kept frozen. 

### 4.3. RNA Sequencing Analysis and Data

We performed RNA sequencing on cortical samples to determine RNA expression changes in the dorsomedial prefrontal cortex related to suicidal behavior. Sample preparation, sequencing library construction and RNA sequencing were conducted by the Beijing Genomics Institute’s (BGI, Hongkong, Shenzhen, China) standard procedure. To extract total RNA, 20 mg of postmortem brain tissue was subjected to RNA sequencing on the BGISEQ-500 platform (BGI) using the 100-bp paired-end sequencing strategy. RNA size, concentration and integrity were verified using Agilent 2100 Bioanalyzer (Agilent Technologies). All the generated raw sequencing reads were filtered using SOAPnuke (1.5.2), a filter software developed by BGI company, to remove reads containing adapters, reads in which unknown bases are more than 5%, and low-quality reads (>20% of the bases with a quality score lower than 15). After filtering, the remaining clean reads were obtained and stored in FASTQ format [133]. HISAT2 (2.0.4) software in combination with the Bowtie index (2.2.5) [134,135] was used to map clean reads to the human reference genome (hg19), respectively. The average mapping ratio with genes was 76.12%, and the average mapping ratio with reference genome was 91.28% (Appendix A). The output of HISAT2 was imported into the RStudio environment (R version 4.0.4; RStudio version 1.4.1106) using the Rsubread package (2.4.3), which converted data from the transcript level to the gene level, then aligned to the human genome (GRCh38) and counted into genes focusing on all annotated protein-coding genes. The Rsubread facilitates the RNA-seq read data analyses, producing the quality assessment of sequence reads, read alignment and read summarization, among others [136]. To identify differentially expressed genes (DEGs), the DESeq2 package (1.34.0) as the standard workflow was used for the detection of DEGs [137]. Before using DESeq2, we performed minimal prefiltering to keep the rows that have at least one read total. To increase power, stricter filtering is automatically applied via independent filtering on the mean of normalized counts within the *results* function. The complete dataset of DEGs is listed in Appendix A. The DEGs were identified by Benjamini and Hochberg-adjusted *p*-value (*p*adj < 0.05) [138] and log2 FC ratio ≥ ±1 was defined as the thresholds to discriminate significant DEGs. DEGs were visualized in a volcano plot created with the R package ggplot2 (3.3.5). Hierarchical clustering of DEGs was analyzed using the *heatmap.2* function in the gplots (3.1.1) package for R. Heatmap created using two color arrays. Pathway-based and co-expression analysis helps to further understand genes’ biological functions.

### 4.4. Gene Ontology and Pathway Enrichment Analysis of DEGs

To assess the function of the DEGs, functional enrichment analysis was performed using the gprofiler2 R package (version 0.2.0). The Gene Ontology (GO) analysis was queried using the *gost* function in “ordered” mode [139]. The differentially expressed genes were ranked by the adjusted *p*-value significance (*p*.adjust < 0.05). Functional enrichment was performed for upregulated and downregulated DEGs, separately. The Kyoto Encyclopedia of Genes and Genomes (KEGG) and Reactome resources were used for functional annotation and pathway analysis [140,141]. The results of the functional enrichment were visualized in Manhattan plots created with the gprofiler2. The significant GO terms were listed in Appendix A. The functional enrichment analyses of the top 10 down- and upregulated DEGs, and also the top 10 down- and upregulated hub genes identified by gene co-expression networks, were carried out using Database for Annotation, Visualization and Integration Discovery (DAVID) v6.8 online server to annotate gene-related biological mechanisms using standardized gene terminology.

### 4.5. Compare RNA-Seq Results with the Allen Human Brain Atlas Database

Cell types in the brain are diverse with broadly different expressional profile, morphology and function. Thus, there is a need to define the functional assessment of DEGs to localize their expression in molecularly defined subtypes of the DMPFC. To address this need, we used the Allen Human Brain Atlas (AHBA) transcriptomic data derived from adult human brain. We used the “Multiple Cortical Area—SMART-seq (2019) dataset for cell-type specific genes which was downloaded from the Allen Institute for Brain Science website (www.brain-map.org, (accessed on 13 December 2021)).

### 4.6. Protein–Protein Interaction Network Construction

As an alternative approach, STRING version 11.5 online tool [142] was used to construct a protein–protein interaction (PPI) network of selected genes. Using the STRING database, genes with a minimum required interaction score of 0.4 were chosen to build a network model visualized by Cytoscape software version 3.8.2 [143]. CytoHubba [144], a Cytoscape Plugin, was used to evaluate the most ranked genes of PPI network. The top 10 most ranked genes were screened out by node degree from both down- and upregulated gene networks. Red colors denoted the higher degree of a gene. Gene list enrichment of the top 10 most ranked DEGs was identified through the stringApp. Functional annotation terms were considered significantly enriched with an FDR-corrected *p*-value < 0.05.

### 4.7. Disease-Associated Gene Sets

Human disease-associated gene sets were obtained from the DisGeNET database, which is known to integrate disease-gene links from several sources [126]. The list of top 10 down- and upregulated DEGs and hub genes were imported into the DisGeNET database browser (http://www.disgenet.org/, (accessed on 15 December 2021)), then the curated gene-disease association file was downloaded. Keywords with special emphasis on mood disorders and associated comorbidities were used for filtering categories to gather those genes involved in the pathophysiology of depression. The result was imported and listed in Appendix A.

### 4.8. Co-Expression Network Construction and Functional Annotation

Coexpression correlation was computed using the Pearson correlation coefficient, as described previously [145]. Read counts of each gene and each sample were normalized via median normalization using the EBSeq R package [146]. The correlation (Pearson’s) and correlation significance of every pair DEG (for downregulated genes adjusted *p*-value < 0.01, for upregulated genes adjusted *p*-value < 0.05) was calculated using a logarithmic matrix of read counts with the psych R package [147]. A network table containing the statistically significant correlations across the whole data set for every pair of DEGs was generated, and to calculate network statistics, igraph R package [148] was used. The network statistics (degree and betweenness centrality for each node) were uploaded to Cytoscape. The gene symbols were designated as the identifiers of nodes. The correlation, degree and betweenness were mapped to the edge color, edge width and node size, respectively. Regarding the topological analysis, GLay community clustering evaluation [149] was performed in order to identify densely connected nodes. The functional annotation analysis of the network was performed using the ClueGO application [150]. Functional annotation terms were considered significantly enriched with a Bonferroni-corrected *p*-value < 0.05. Further details on the interplay of the reconstructed networks were examined using the stringApp. For screening hub genes from co-expression networks, a Maximal Clique Centrality (MCC) algorithm was used to select the hub genes [151]. The network datasets and functional annotation results of the networks were shown in Appendix A.

### 4.9. Validation of Expression Changes by qRT-PCR

To confirm the RNA-seq data, a subset of differentially expressed genes whose selection was based on the known or potentially probable involvement in depression and neuron-specific functions were validated using quantitative real-time PCR (qRT-PCR). The procedure was carried out as described previously [152]. Briefly, total RNA was isolated from approximately 20 mg of frozen postmortem brain tissue—from the same samples used in RNA-seq—using TRIzol reagent (Invitrogen, Carlsbad, CA, USA) as lysis buffer combined with RNeasy Mini kit (Qiagen, Germany) following the manufacturer’s instructions. The quality and quantity of extracted RNA were determined using NanoDrop ND-1000 Spectrophotometer (Thermo Fisher Scientific, Waltham, MA, USA), and only those with A260/A280 ratio between 1.8 and 2.1 were used in subsequent experiments. The concentration of RNA was adjusted to 500 ng/µL, and it was treated with Amplification Grade DNase I (Invitrogen, Carlsbad, CA, USA). The isolated RNA concentration was calculated and normalized with RNase-free water and reverse transcribed into cDNA using SuperScript II Reverse Transcriptase Kit (Invitrogen, Carlsbad, CA, USA). After 10-fold dilution, 2.5 μL of the resulting cDNA was used as a template in PCR performed in duplicate using SYBR Green dye (Sigma, St Louis, MO, USA). The PCR reactions were performed on CFX-96 C1000 Touch Real-Time System (Bio-Rad Laboratories, Hercules, CA, USA) with iTaq DNA polymerase (Bio-Rad Laboratories, Hercules, CA, USA) in total volumes of 12.5 μL under the following conditions: 95 °C for 3 min, followed by 35 cycles of 95 °C for 0.5 min, 60 °C for 0.5 min and 72 °C for 1 min. A melting curve was performed at the end of amplification cycles to verify the specificity of the PCR products. The primers used for qRT-PCR were synthesized by Integrated DNA Technologies, Inc., (IDT, Coralville, IA, USA) and used at 300 nM final concentration. Sequences of primers are listed in Appendix A. Housekeeping genes ACTB, GAPDH and LDHA were used as internal controls and the relative gene expression values were calculated from their averages using the 2^−^^△△Ct^ method. 

### 4.10. Preparation of In Situ Hybridization Probes

A mixture of cDNAs prepared with RT-PCR from the human DMPFC was used as a template in PCR reactions with primers for NECAB2 (CAGGATCTTGGTGCCAGCT and TGTGGTCAGTGTGGGTCATG) yielding a probe that corresponds to the GenBank accession number NM_019065.2. The purified PCR products were applied as templates in a PCR reaction with the primer pairs specific for the probe and also contained the T7 RNA polymerase recognition site added to the reverse primers. Finally, the identities of the cDNA probes were verified by sequencing them with T7 primers.

### 4.11. In Situ Hybridization Histochemistry

Two freshly frozen DMPFC brain blocks of subjects were used: one from a 75-year-old female control subject with negative clinical reports for any major diseases and one from a 72-year-old female suicidal individual. Using a cryostat, serial coronal sections (12 μm) were cut and mounted on positively charged slides (SuperfrostPlus^®^, Fisher Scientific), dried and stored at −80 °C until use. Further steps were performed according to the procedure described previously by Dobolyi et al. [153]. Briefly, antisense [35S]UTP-labeled riboprobes were generated from the above-described DNA probes using T7 RNA polymerase of the MAXIscript Transcription Kit (Ambion, Austin, TX, USA) and used for hybridization at 1 million DPM (discharges per minute) activity per slide. Washing procedures included a 30 min incubation in RNase A, followed by decreasing concentrations of sodium-citrate buffer (pH 7.4) at room temperature and subsequently at 65 °C. Following hybridization and washes, slides were dipped in NTB nuclear track emulsion (Eastman Kodak) and stored at 4 °C for 3 weeks for autoradiography. Then, the slides were developed and fixed with Kodak Dektol developer and Kodak fixer, respectively, counterstained with Giemsa and coverslipped with Cytoseal 60 (Stephens Scientific, Kalamazoo, MI, USA).

### 4.12. Tissue Collection for Immunolabeling

Immunohistochemistry was used to assess the distribution of NECAB2 protein in the DMPFC. For immunolabeling, one DMPFC brain block from a 62-year-old female control individual was cut into 10 mm thick coronal slice and immersion fixed in 4% paraformaldehyde in 0.1 M phosphate-buffered saline (PBS) for 5 days. Subsequently, the block was transferred to PBS containing 0.1% sodium azide for 2 days to remove excess paraformaldehyde. Then, the block was placed in PBS containing 20% sucrose for 2 days of cryoprotection. The block was frozen and cut into 60 μm thick serial coronal sections on a sliding microtome. Sections were collected in PBS containing 0.1% sodium azide and stored at 4 °C until further processing.

### 4.13. DAB Immunolabeling

Free-floating DMPFC brain sections were immunolabeled for NECAB2 (Thermo Fisher Scientific, Cat. No. PA5-53108). The antibody (1:250 dilution) was applied for 24 h at room temperature, followed by incubation of the sections in biotinylated anti-rabbit secondary antibody (1:1000 dilution, Vector Laboratories, Burlingame, CA, USA) and then in avidin–biotin–peroxidase complex (1:500, Vector Laboratories) for 2 h. Subsequently, the labeling was visualized via incubation in 0.02% 3,3-diaminobenzidine (DAB; Sigma), 0.08% nickel (II) sulfate and 0.001% hydrogen peroxide in PBS, pH 7.4 for 5 min. Sections were mounted, dehydrated and coverslipped with Cytoseal 60 (Stephens Scientific, Riverdale, NJ, USA).

### 4.14. Double Labeling of NECAB2

Double immunofluorescence staining was used to clarify the colocalization of NECAB2 in the DMPFC. To reduce autofluorescence, tissue sections were treated with 0.15% Sudan Black B (in 70% ethanol) after antigen retrieval (0.05 M Tris buffer, pH = 9.0) procedures. Slides were blocked by incubation in 3% bovine serum albumin (with 0.5% Triton X-100 dissolved in 0.1 M PB, Sigma) for 1 h at room temperature, followed by washing with washing buffer (10 min × 3). NECAB2 was immunolabeled for single labeling using 1:250 dilution except for the visualization, which was performed with fluorescein isothiocyanate (FITC)-tyramide (1:8000 dilution) and H_2_O_2_ (0.003%) in 100 mM Trizma buffer (pH 8.0 adjusted with HCl) for 6 min. Subsequently, sections were placed in goat anti-ionized calcium-binding adapter molecule 1 (Iba1) (1:500 dilution, Abcam, Cat. No. ab107159), and mouse anti-S100 (1:250 dilution, Millipore, Cat. No. MAB079-1) for 24 h at room temperature. The sections were then incubated in Alexa 594 donkey anti-goat/mouse secondary antibody (1:500 dilution, Vector Laboratories) followed by 2 h incubation in a solution containing avidin–biotin–peroxidase complex (ABC, 1:300 dilution, Vector Laboratories). Finally, all sections with fluorescent labels were mounted on positively charged slides (Superfrost Plus, Fisher Scientific, Pittsburgh, PA, USA) and coverslipped in antifade medium (Prolong Antifade Kit, Molecular Probes).

### 4.15. Microscopy and Photography

Sections were examined using an Olympus BX60 light microscope also equipped with fluorescent epi-illumination and a dark-field condenser. Images were captured at 2048 × 2048 pixel resolution with a SPOT Xplorer digital CCD camera (Diagnostic Instruments, Sterling Heights, MI, USA) using a 4× objective for dark-field images, and 4–40X objectives for bright-field and fluorescent images. Confocal images were acquired with a Zeiss LSM 70 Confocal Microscope System using a 40-63X objectives at an optical thickness of 1 µm to count varicosities and 3 µm to count labeled cell bodies. Contrast and sharpness of the images were adjusted using the ‘levels’ and ‘sharpness’ commands in Adobe Photoshop CS 8.0. Full resolution was maintained until the photomicrographs were cropped, at which point the images were adjusted to a resolution of at least 300 dpi.

### 4.16. Statistical Analysis

Demographics were contrasted using the Chi-Square test or Welch’s unequal variances *t*-test (*p* < 0.05). The plot and heatmap were generated using R version 4.0.4 (https://www.r-project.org/, (accessed on 10 January 2022)). Quantitative statistics of qPCR results were performed with the GraphPad Prism version 8.0.1 (GraphPad Software, San Diego, CA, USA). Normality distribution was assessed via Shapiro–Wilk test. For comparisons between values of the two groups, Welch’s unequal variances *t*-test was applied. The nonparametric Mann–Whitney U test was used for data that were not normally distributed. Differences were considered statistically significant when *p* < 0.05 and plotted as mean ± S.E.M (standard error of the mean).

## 5. Conclusions

We performed a comprehensive transcriptome study of DMPFC in suicide victims, using RNA-seq and observed significant associations between specific transcriptome changes in DMPFC and suicidal behavior. They include (but are not limited to) the suppression of genes that regulate the inactivation of immune responses and glial cell differentiation, and the upregulation of genes involved in glutamatergic synaptic transmission. The identified gene expression alterations may reveal dysregulated pathways and functional cascades involved in the pathophysiology of suicidal behavior. Therefore, we conclude that the DEGs and pathways identified in this study could be linked to depression, albeit a common secondary pathology arising from dysregulated network activity is also possible. These results are supported by the large overlap of DEGs between previously reported genes altered in depressive disorders. Furthermore, due to the alteration in pathways relating to gliogenesis, suicidal behavior can affect glial cell numbers and cellular morphology in the medial part of the prefrontal cortex. Additionally, alterations in glutamate neurotransmission including both ionotropic and metabotropic receptors, as well as neuroinflammatory processes potentially contributing to neuroplastic changes, can contribute to the dysfunction of the DMPFC. The DMPFC is one of the nodes in connections with different cortical networks receiving direct viscero- and somatosensory inputs, motivational/emotional inputs from the orbitofrontal cortex and cognitive inputs from the dorsolateral and ventral prefrontal cortex. By processing all this information together using activated working memories, the DMPFC may work out motor programs and projects (through the default mode network) to the precuneus and directly to the premotor and supplementary motor cortical areas. The precuneus is able to interfere or even block different actions of the motor cortex, and through self-reflection may determine suicidal behavior.

Taken together, our findings may contribute to the pathophysiology of depressive or suicidal behavior. The high percentage of altered genes in suicide victims involved in depression is in line with the fact that depression is the leading cause of suicide. We conclude that our study provides new insights into the genomic underpinnings of major depressive disorder and may allow for novel precision therapeutic development strategies targeting depression to be implemented in future studies.

## Figures and Tables

**Figure 1 ijms-23-07067-f001:**
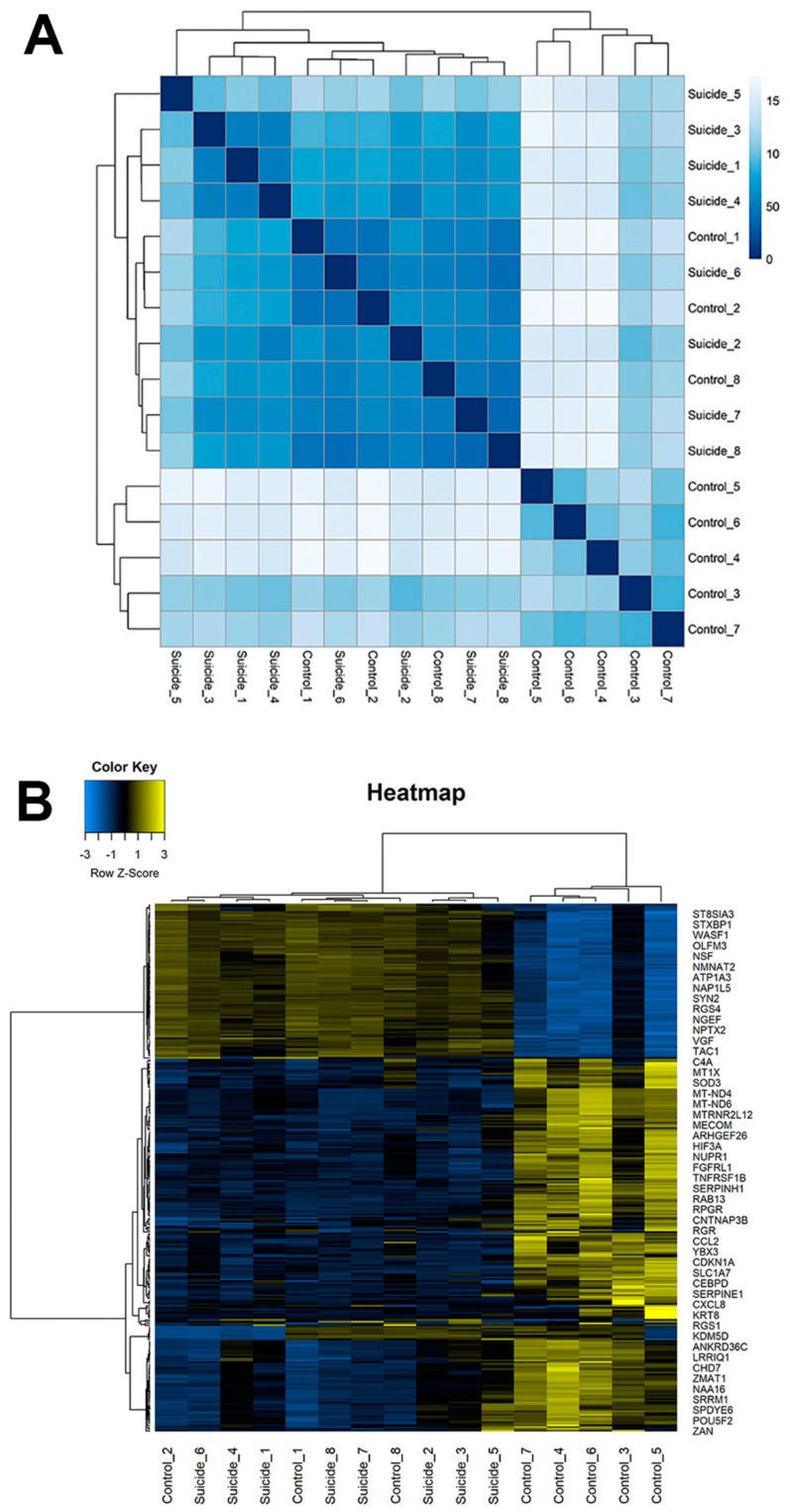
Analysis of gene expression between suicide and control groups. (**A**) Hierarchical clustering of 16 RNA-seq samples based on the Euclidean distance. A heatmap of the distance matrix shows the similarities and dissimilarities between the samples as calculated from the rlog transformation of the count data. (**B**) Hierarchical clustering heatmap of Pearson correlation coefficients between suicide and control individuals. The heatmap represents the top 500 protein-coding gene expression for the different groups in columns, with a dendrogram presented at the top of the heatmap. Proteins were selected based on their variability between all samples. Counts were log-transformed, normalized, and used for clustering based on similarity in expression patterns. The x-axis represents the sample. The y-axis represents the top 500 differentially expressed genes. The color represents the log10 transformed gene expression level. The scale shows the level of expression: the yellow color means high, the blue color means low, while the black color represents medium expression level.

**Figure 2 ijms-23-07067-f002:**
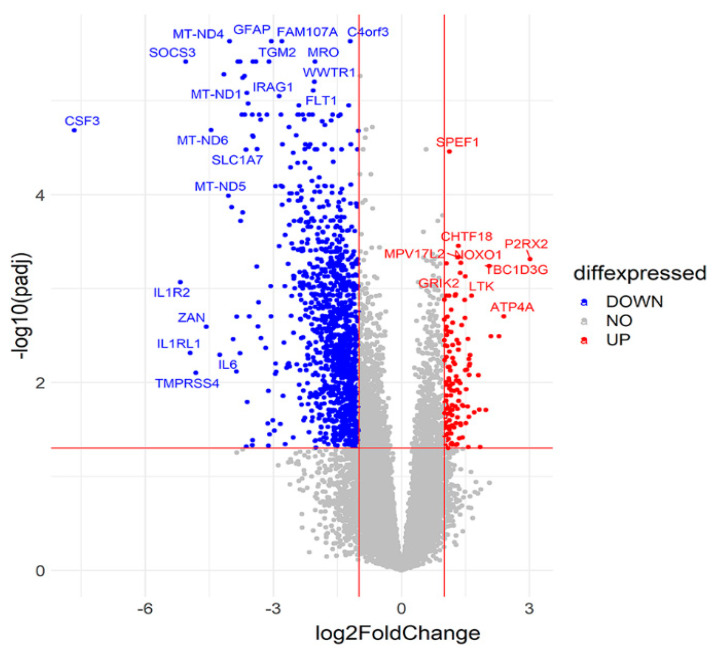
Visualization of RNA-seq results. Volcano plot showing the log2 fold change (log2FC) of gene expression and the statistical significance of the differential expression (DE) analysis performed between suicide and control individuals. The x-axis represents the log2 fold change of genes, while the y-axis represents the −log10 of the corrected *p*-values (padj) for the different pairs of conditions. Each dot represents a gene and the colored areas represent the DEGs that met the following selection criteria: log2FC of at least ±1 (log2FC ≥ 1 or ≤−1) and Benjamini and Hochberg-adjusted *p*-value < 0.05. Upregulated genes are shown in red, while the downregulated ones are blue. The top 40 significant DEGs are labeled.

**Figure 3 ijms-23-07067-f003:**
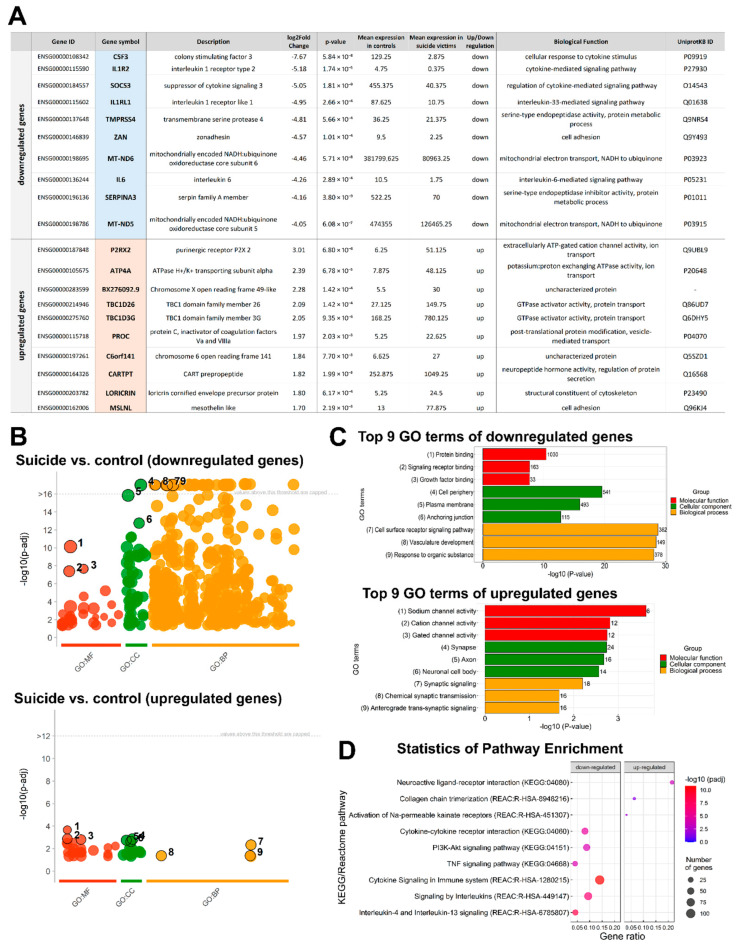
Functional enrichment of differentially expressed genes. (**A**) Top 10 down- and upregulated differentially expressed genes ranked by the value of the log2 fold change (log2FC) and their biological functions. (**B**) Manhattan plot shows the enrichment results of down- and upregulated genes. The x-axis shows the terms and the y-axis shows the enrichment *p*-values on the −log10 scale. Each circle on the plot corresponds to a single term. Circles are colored according to the origin of annotation and size scaled according to the total number of genes annotated to the corresponding term. The locations on the x-axis are fixed. Terms from the same GO subtree are located closer to each other on the x-axis, which helps to highlight different enriched GO sub-branches making plots from different queries comparable. The top 3 terms from each GO category are indicated with numbers on the plot. The corresponding statistics of annotation are listed in Appendix A. (**C**) GO classification and scatter plot-enriched Kyoto Encyclopedia of Genes and Genomes (KEGG) pathways of DEGs. Bar plots reporting the top 9 significantly enriched GO terms related to down- and upregulated genes with the adjusted *p*-value < 0.05, respectively. The table describes, for each GO term, the number of mapped annotated genes in the reference data set and its −log10 *p*-value. The color code is proportional to the three ontologies. (**D**) Dot plots of top 3 pathway annotations from KEGG and Reactome pathways illustrate the distributions of gene sets among up- and downregulated differentially expressed genes. The x-axis represents the gene ratio. The y-axis represents the pathway term. Gene ratio refers to the value of enrichment, which is the ratio of DEGs annotated in the pathway to the total gene amount annotated in the pathway. The larger the value, the more significant the enrichment. The circle size indicates the DEG number associated with each significant pathway. The color indicates the adjusted *p*-value, the lower *p*-value indicates the more significant enrichment.

**Figure 4 ijms-23-07067-f004:**
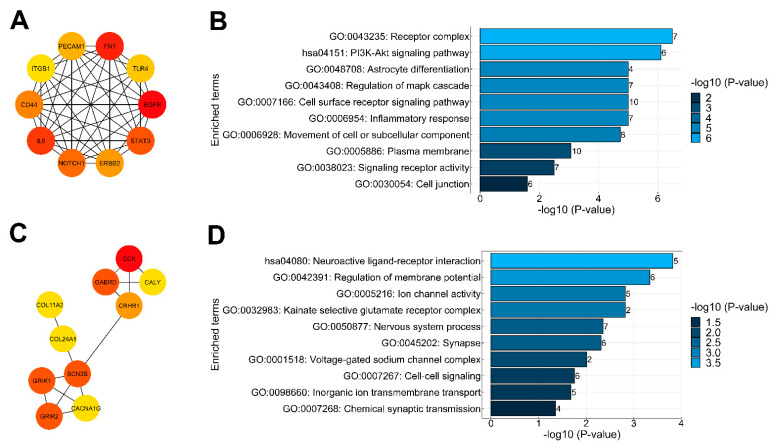
The top 10 down- and upregulated DEGs of PPI network. (**A**,**C**) The top 10 ranked DEGs of PPI networks were obtained from CytoHubba analysis based on the degree method. (**A**) The panel shows the top 10 DEGs of downregulated genes and (**C**) represents the top 10 DEGs of upregulated genes. The change in color from red to yellow represents a change in the degree score from high to low. (**B**,**D**) Gene Ontology (GO) functional and KEGG pathway classification of top 10 DEGs analyzed through stringApp with the FDR-corrected *p*-value < 0.05. (**B**) The panel shows the enriched terms of top 10 downregulated DEGs, while (**D**) shows enriched terms of top 10 upregulated DEGs. On the graph, y-axis represents the significant enrichment terms. Each bar describes the number of mapped annotated genes in the reference data set while the x-axis and color gradient indicate the significance (−log10 *p*-value) in each category.

**Figure 5 ijms-23-07067-f005:**
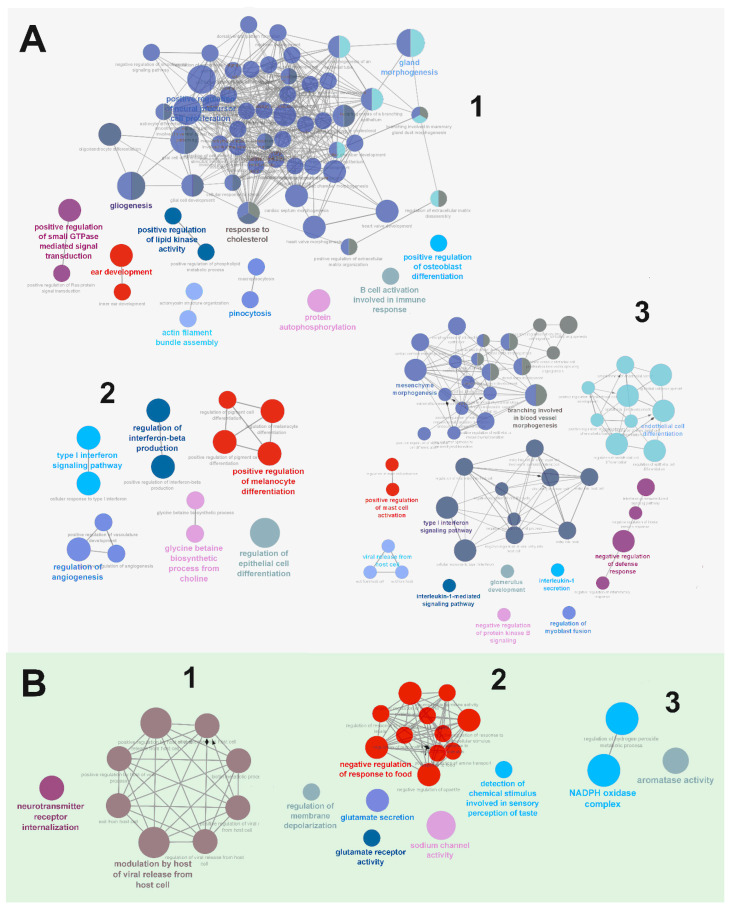
Co-expression network analysis and functional annotation of DEGs. Visualization of statistically overrepresented GO terms in a given set of genes from the three main co-expression network clusters (1–3) using the ClueGO plugin. Functional annotation of DEGs using ClueGO enriched by downregulated (**A**) and upregulated DEGs (**B**). Each node refers to a specific Gene Ontology (GO) Biological process term and is grouped based on the similarity of their associated genes. The size of the nodes is equivalent to the statistical power of significance of each term: the larger the node size is, the smaller the *p*-value is. Node colors represent different functional groups. The emphasized term of each functional group is given by the most significant term of the group.

**Figure 6 ijms-23-07067-f006:**
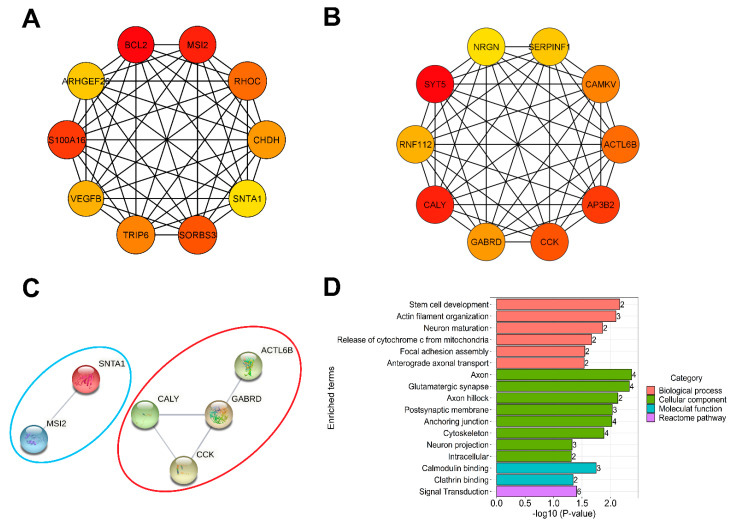
Hub gene analysis of co-expression networks. (**A**,**B**) The top 10 hub genes from down- and upregulated co-expression networks were screened according to the MCC score using the CytoHubba plugin of Cytoscape. (**A**) shows the top 10 hub genes of downregulated co-expression network and (**B**) represents the top 10 hub genes of upregulated co-expression network. The change in color from red to yellow represents a change in degree score from high to low. (**C**) PPI network of hub genes constructed using STRING Online Database. The network contains 20 nodes and 5 edges compared to the expected number of edges (1). The PPI enrichment *p*-value was 0.00815. The blue circle represents the interaction network of downregulated hub gens, while the red circle indicates the interaction for upregulated hub genes. (**D**) GO and Reactome pathway classification of the top 10 down- and upregulated hub genes. Bar plots describing the significantly enriched terms related to down- and upregulated hub genes with the *p*-value < 0.05. The table describes, for each term, the number of mapped annotated genes in the reference data set and its −log10 *p*-value. The color code is proportional to the ontologies.

**Figure 7 ijms-23-07067-f007:**
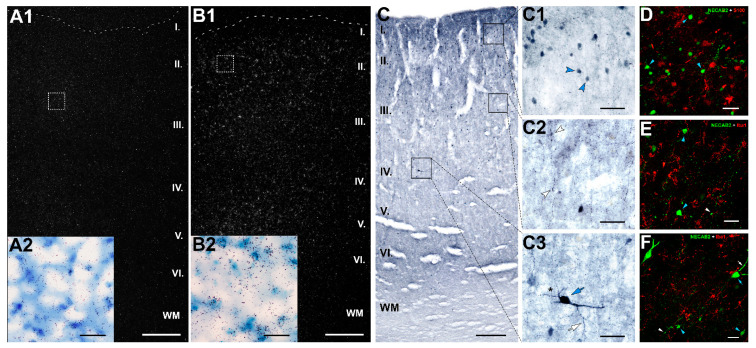
Characterization of NECAB2-immunoreactive (NECAB2-ir) neurons in the DMPFC. (**A**,**B**) NECAB2 mRNA expression in the DMPFC of control (**A1**) and suicide (**B1**) individuals were visualized by in situ hybridization histochemistry. A dark-field photomicrograph shows high intensity of NECAB2 hybridization signal in the DMPFC. NECAB2 signal is prominent in cortical layers II and IV in the DMPFC of suicide victims (**B1**) compared to the representative control (**A1**). High magnification bright-field microphotographs of tissue sections (**A2**,**B2**) show the area assigned by the dashed boxes in (**A1**,**B1**) and developed for silver grains in the control (**A2**) and suicide (**B2**) victims. Cortical layers are indicated on the left. (**C**) Laminar distribution of NECAB2-ir neurons across cortical layers in the left DMPFC of a female control individual (62 years old) visualized by DAB immunostaining. NECAB2-ir neurons are mainly located in II–V layers and as shown on (**C1**,**C3**), NECAB2 is located in at least 2 different interneuron subtypes based on the shape and size of labeled neurons. (**C1**–**C3**) Higher magnification of the boxed area in (**C**). (**C1**) NECAB2-ir neurons are presented in cortical layer II of the DMPFC (blue arrowhead). (**C2**) NECAB2-ir axons give rise to varicosities (white arrowheads) along the section. (**C3**) A higher magnification photomicrograph of a layer IV NECAB2-immunopositive interneuron shows the soma (blue arrow), the varicose dendritic tree (asterisk) and a part of the axon (white arrow). Cortical layers are indicated on the left. (**D**–**F**) Representative confocal microscopy images of double immunolabeling of Iba1 and S100B glial markers (red) with NECAB2 (green) in DMPFC control sections. (**D**) High-magnification confocal image of a section double-labeled with S100B (red) indicates that NECAB2 (green) does not colocalize with S100-positive astrocytes. (**E**,**F**) Likewise, NECAB2-positive cells and Iba1-positive microglia do not have overlapping distributions. Scale bar represents 500 µm in (**A1**,**B1**), 250 µm in (**C**), and 50 µm in (**A2,B2**,**C1**–**C3**,**D**–**F**). Abbreviations: I-VI–cortical layers; WM–white matter.

**Figure 8 ijms-23-07067-f008:**
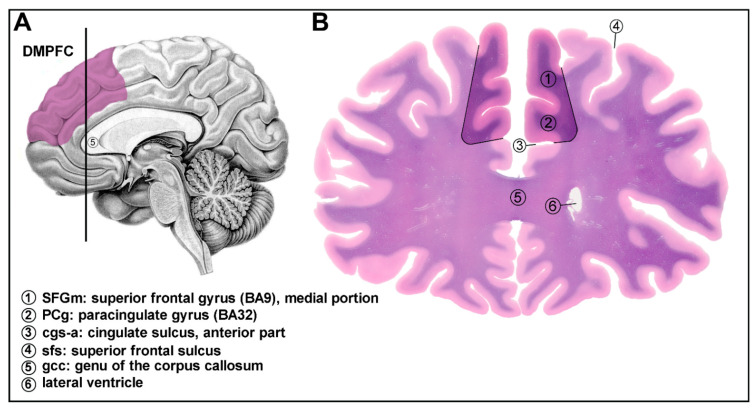
A representative view of the human brain region was dissected in the study. Details on the anatomical origin and structure of the dissected region. (**A**) The location and topographical extension of the dissected area, the dorsomedial prefrontal cortex (DMPFC) is shown in lilac color on the medial surface of the human brain. The vertical black line represents the cutting line used to obtain the coronal section illustrated on the right panel. (**B**) Coronal section of the human brain represents the DMPFC (1 + 2). The dissected area analyzed in this study is highlighted and demarcated by black lines. Landmarks and boundaries are notified by numbers. The section was stained using the Levanol-Fast Cyanine 5RN method.

**Table 1 ijms-23-07067-t001:** Validation of DEGs with qRT-PCR. Based on the RNA-seq analysis, 14 differentially expressed genes were selected. For validation, quantitative real-time polymerase chain reaction (qRT-PCR) was used, and the results showed significantly altered expression of all genes in the expected direction based on RNA-seq.

Gene ID	Gene Symbol	Description	log2 Fold Change	*p*-Value	Up/Down Regulation	qPCR log2 Fold Change	qPCR *p* Value	mRNA Expression in Controls (Mean ± SEM)	mRNA Expression in Suicide Victims (Mean ± SEM)	Function
ENSG00000164418	GRIK2	glutamate ionotropic receptor kainate type subunit 2	1.37	1.22 × 10^−5^	up	2.35	0.007	0.002 ± 0.001	0.005 ± 0.001	kainate selective glutamate receptor activity
ENSG00000154146	NRGN	neurogranin	1.34	7.44 × 10^−4^	up	2.16	0.005	0.083 ± 0.029	0.247 ± 0.032	signal transduction
ENSG00000129990	SYT5	synaptotagmin 5	1.26	1.77 × 10^−3^	up	3.76	0.038	0.001 ± 0.0005	0.002 ± 0.0004	SNARE, syntaxin binding
ENSG00000164082	GRM2	glutamate metabotropic receptor 2	1.19	1.70 × 10^−3^	up	2.18	0.021	0.0003 ± 0.0001	0.001 ± 0.0001	adenylate cyclase inhibiting G protein-coupled glutamate receptor activity
ENSG00000171189	GRIK1	glutamate ionotropic receptor kainate type subunit 1	1.16	5.26 × 10^−5^	up	1.59	0.021	0.007 ± 0.002	0.016 ± 0.002	kainate selective glutamate receptor activity
ENSG00000103154	NECAB2	N-terminal EF-hand calcium-binding protein 2	1.00	7.82 × 10^−5^	up	1.05	0.037	0.022 ± 0.005	0.038 ± 0.004	type 5 metabotropic glutamate receptor binding
ENSG00000240583	AQP1	aquaporin 1	−2.59	2.29 × 10^−7^	down	−1.72	0.045	0.003 ± 0.001	0.001 ± 0.0002	transmembrane transporter activity
ENSG00000143772	ITPKB	inositol-trisphosphate 3-kinase B	−2.40	1.55 × 10^−8^	down	−1.57	0.036	0.019 ± 0.005	0.005 ± 0.001	ATP binding, kinase activity
ENSG00000132470	ITGB4	integrin subunit beta 4	−2.26	9.70 × 10^−7^	down	−2.1	0.034	0.004 ± 0.001	0.001 ± 0.0001	G protein-coupled receptor binding
ENSG00000137491	SLCO2B1	solute carrier organic anion transporter family member 2B1	−1.79	9.99 × 10^−8^	down	−2.05	0.048	0.032 ± 0.012	0.004 ± 0.0004	sodium-independent organic anion transmembrane transporter activity
ENSG00000152661	GJA1	gap junction protein alpha 1	−1.75	2.64 × 10^−4^	down	−0.78	0.049	0.054 ± 0.009	0.032 ± 0.004	glutathione transmembrane transporter activity
ENSG00000027075	PRKCH	protein kinase C eta	−1.71	8.74 × 10^−7^	down	−2.11	0.047	0.01 ± 0.003	0.001 ± 0.0001	calcium-dependent protein kinase C activity
ENSG00000135821	GLUL	glutamate-ammonia ligase	−1.51	3.07 × 10^−4^	down	−1.78	0.039	8.703 ± 2.713	1.807 ± 0.252	glutamate-ammonia ligase activity
ENSG00000160307	S100B	S100 calcium-binding protein B	−1.39	5.78 × 10^−6^	down	−1.11	0.037	0.528 ± 0.128	0.198 ± 0.01	calcium-dependent protein binding

The genes are arranged according to their fold-change values (log2FC).

**Table 2 ijms-23-07067-t002:** Demonstration of down- and upregulated DEGs, which share common functions with depression-related pathways.

Downregulated DEGs	Upregulated DEGs	Function	Depression-Related Pathway
GABRG1, NTSR2, GPR37L1, GABRE, GLRA1, GRIN2C	CARTPT, GABRD, CCK, CRHR1, GRM2, GRIK1, GRIK2	Signal transduction	Neuroactive ligand-receptor interaction
ITPR2, RYR3, ASPH	TRPM2	Calcium-mediated signaling	Oxytocin signaling pathway
SLC6A13, SLC6A11, SLC6A12, GABRG1, GABRE, SLC38A3, SLC38A5	GABRD	Anion transmembrane transporter activity	GABAergic synapse
PDGFRB, FLT4, ERBB2, EGFR, NTRK2, CYSLTR2, FLT1, PTGER1, GRIN2C, NOS3, ADORA2B, VEGFB, ADCY4, FGF8, ADORA2A, FGFR3, GNA14, ASPH, ITPR2, RYR3, ITPKB, PLCD1, TPCN1, PLCD3, FGF1	CACNA1G, FGF8, P2RX2	Cell communication	Serotonergic synapse
TNFRSF1A, TGFB1, TGFB3, DUSP1, RRAS, HSPB1, PDGFRB, FLT4, ERBB2, GNA12, EGFR, FLT1, VEGFB, ANGPT2, CSF1, MAP4K4, EPHA2, TGFBR2, TGFB2, IL1R1, PGF, FGF1	CACNA1G, CACNG8, DUSP4, FGF8	Regulation of cellular process	MAPK signaling pathway

**Table 3 ijms-23-07067-t003:** Demographic data of individuals.

Donor	Sex	Age	Post Mortem Interval (PMI)	Cause of Death	Clinical and Pathological Diagnosis
#1	female	48	6–7 h	Suicide (drug overdose)	-
#2	male	71	1 h	Suicide (jumping from a height)	Without any clinical care during the past 6 months
#3	male	48	6 h	Suicide (hanging—asphyxia)	Without known drug treatment
#4	female	65	5 h	Suicide (hanging—asphyxia)	Pathological diagnosis: negative status (no pathological sign for any diseases)
#5	male	31	8 h	Suicide (hanging—asphyxia)	Without known drug treatment
#6	female	49	6 h	Suicide (drug overdose)	Without known drug treatment
#7	male	43	4 h	Suicide (hanging)	Without any clinical care
#8	male	66	8–10 h	Suicide (hanging—asphyxia)	Laboratory test: alcohol: negative
#9	male	42	3.5 h	Acute respiratory insufficiency	-
#10	female	56	6 h	Cardiorespiratory insufficiency, edema cerebri	Edema cerebri, coarctatio aortae, hepatitis alcoholica
#11	male	50	5.5 h	Stroke, brain hemorrhage	Large cortical and subcortical hemorrhage in the parietal lobe
#12	male	68	10 h	Acute heart failure	Acute pulmonary edema, serious arteriosclerosis (especially in the heart and kidney), peripheral arterial shunt, cerebral sclerosis. left coronary occlusion
#13	female	75	10 h	Stroke (right side arteria cerebri media)	Diabetes, stroke, hypertonia, mamma carcinoma, emolitio arteriae cerebri mediae lateralis dextri, cortical infarction, general atherosclerosis
#14	male	64	10 h	Stroke (arteria cerebri media on the left side), bronchopneumonia	Cardiomyopathia, coronary sclerosis, hypertonia, infarctus myocardii, bronchopneumonia, cardiorespiratory insufficiency, femoralis amputatio, aphasia, carotis stenosis, pneumonia
#15	male	90	4–5 h	Stroke (cerebri media and posterior)	Stroke, infarctus lacunaris multiplex cerebri, Parkinson’s disease, emolitio, tracheobronchitis, cardiopulmonary insufficiency, carotis stenosis
#16	male	78	10 h	Cardiorespiratory insufficiency	Dementia, diabetes, hypertonia, carotis interna occlusio, polyneuropathia

## Data Availability

The data presented in this study are available in the Appendix A of this article. The raw sequencing data files are available in the Sequence Read Archive (SRA) at the National Center for Biotechnology Information (NCBI) under the Bioproject ID: PRJNA828151, with the following sample accession numbers: SAMN27632065, SAMN27632062, SAMN27632064, SAMN27632063, SAMN27632060, SAMN27632061, SAMN27632059, SAMN27632058, SAMN27632071, SAMN27632070, SAMN27632068, SAMN27632069, SAMN27632066, SAMN27632057, SAMN27632056 and SAMN27632067.

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
