# Peer review of "Transcriptome Profiling of the Dorsomedial Prefrontal Cortex in Suicide Victims"

_ijms, 2022, doi:10.3390/ijms23137067_

Round 1

Reviewer 1 Report

The manuscript by Dóra et al does a whole tissue transcriptomic analysis of dorsomedial prefrontal cortex samples from human subjects who died from suicides vs natural death to find suicide-specific gene expression changes. The DMPFC is an understudied region in the context of suicidal behavioral related pathology, and the authors’ choice is novel. The authors find upwards of 1000 differentially expressed genes which is par for the field. They then perform several types of bioinformatic analyses to pull out the different molecular pathways that these DEGs participate in. Broadly, their analyses reveals that inflammation-related signaling and gliogenesis modules were negatively regulated and glutamatergic signaling modules were upregulated in brain tissue from suicide victims. The authors do a good job of comparing their observations against available literature for gene expression changes in MDD and suicide.

But the reviewer has some concerns with the manuscript in its current form. They are listed below:

MAJOR POINTS:

1.       Fig 1: Can the authors comment on why the three controls clustered with the suicide victim samples? Is there a systematic regressible variable that may explain this discrepancy such as age, sex, PMI, sample quality etc?

2.       Section 2.4 and Fig 4: The reviewer does not clearly follow the logic for calculating hubness features for  the top 10 down- or up-regulated genes specifically (ln 199-200). The hubness of a gene or genes should be calculated by running a co-expression analysis against all the genes that were differentially modulated in a given direction (positive or negative), and then pulling out the top ones based on their centrality measurements. The top DEGs would look like hubs if they were the only ones to be tested. The PPI network analysis should be carried out on the hubs identified from the complete list of DEGs. Please clarify this section.

3.       Table 1 & Supp Fig 2: The qRT-PCR results for validating the RNAseq data should have a scatter accompanying the bar graphs (and errors in table 1)6. This would allow readers to appreciate the extent of variability across subjects.

4.       Section 2.7 ln 253-254: What were the reasons for arriving at the criteria for selecting genes with correlation > 0.9 and Padj < 0.01? Were other values in the parameter space tried? It is also not clear what the authors mean by correlation here. What were the genes correlated with?

5.       Ln 270-375: Please indicate the references for the human and rodent studies that are congruent with the DEG/qRT-PCR expression patterns observed in the dataset.

6.       While it is informative to read about the list of changes that the authors reviewed in the discussion, the reviewer feels that a short paragraph describing how they think these pathways could alter DMPFC network output leading to depression and/or suicide will tremendously help the readers. Without such a model, the discussion just reads like a list of changes. While such a model would be highly speculative, it will nevertheless create a more easily digestible message from the paper

7.       Ln 751-755: Considering the heterogeneity of cell types that the samples were collected from, authors should not make claims about causation. While the reviewer agrees that the authors data has several commonalities with other studies looking at expression changes in MDD and suicide from other brain areas, this could just be a common secondary pathology arising from dysregulated network activity.  

8.       Supp Table 1: Considering the wide variety of pathologies (including brain pathologies) that the “control subjects” displayed compared to the suicide victims, can the authors discuss the effects these pathological changes may have had on gene expression. Some of the pathologies listed for the controls are chronic and would affect function and gene expression across brain changes. On the other hand, the suicide victim tissue is largely free of any pathological changes.

MINOR POINTS:

1.       Ln 334-335: The sentence is incomplete.

Author Response

REVIEWER 1.

We appreciate the thorough review, which gave us an opportunity to address several major points to improve the manuscript. Please, find our point-by-point responses below:

MAJOR POINTS:

  1. Fig 1: Can the authors comment on why the three controls clustered with the suicide victim samples? Is there a systematic regressible variable that may explain this discrepancy such as age, sex, PMI, sample quality etc?

We agree with the Reviewer that it is an important point. Therefore, the manuscript was supplemented with the following sentences: “We compared the age, sex and PMI between the three controls clustered with the suicide victims and those who were not clustered using t-test. We found no significant differences between the 2 groups of control individuals for any of these covariates (age: p=0.51; PMI: p=0.39; sex-ratio: p=0.67). Therefore, we do not assume any systematic confounding variables that can distort the results.” Consequently, it is hard to know the reason for the discrepancy except for citing the high variety of human individuals.

  1. Section 2.4 and Fig 4: The reviewer does not clearly follow the logic for calculating hubness features for the top 10 down- or up-regulated genes specifically (ln 199-200). The hubness of a gene or genes should be calculated by running a co-expression analysis against all the genes that were differentially modulated in a given direction (positive or negative), and then pulling out the top ones based on their centrality measurements. The top DEGs would look like hubs if they were the only ones to be tested. The PPI network analysis should be carried out on the hubs identified from the complete list of DEGs. Please clarify this section.

We agree with the reviewer that our original analysis was not appropriate to evaluate hubbness. Therefore, we performed 2 co-expression analyses against all the genes that were up- and downregulated. The new figure showing these results is Fig. 5. Then, we determined the top 10 hub genes of down- and upregulated co-expression networks according to their highest Maximal Clique Centrality (MCC) scores using the CytoHubba plugin in Cytoscape. The new figure showing these results in Figure 5. The description of the results is in lines 312-321.

  1. Table 1 & Supp Fig 2: The qRT-PCR results for validating the RNAseq data should have a scatter accompanying the bar graphs (and errors in table 1)6. This would allow readers to appreciate the extent of variability across subjects.

The figure showing qRT-PCR results was altered so that it shows the scatter of the data in the graphs in order to appreciate individual data points and their variations. Accordingly, Table 1 was supplemented with original data in both groups and Supplementary figure 2 was also modified.

  1. Section 2.7 ln 253-254: What were the reasons for arriving at the criteria for selecting genes with correlation > 0.9 and Padj < 0.01? Were other values in the parameter space tried? It is also not clear what the authors mean by correlation here. What were the genes correlated with?

Our purpose was to identify the highly correlated gene pairs to increase the credibility of our co-expression network. Therefore, we set highly stringent parameters to filter out less correlated gene pairs. In particular, there are several studies where similar criteria were used successfully. Please, find these references below. The above specified part (lines 260-266 in the revised manuscript) has been corrected in the manuscript to better describe our intentions and the way we performed the analysis.

References:

Li C, Cao F, Li S, Huang S, Li W, Abumaria N. Profiling and Co-expression Network Analysis of Learned Helplessness Regulated mRNAs and lncRNAs in the Mouse Hippocampus. Front Mol Neurosci. 2018;10:454. Published 2018 Jan 11. doi:10.3389/fnmol.2017.00454

Tian Y, Xu Y, Wang H, et al. Comprehensive analysis of microarray expression profiles of circRNAs and lncRNAs with associated co-expression networks in human colorectal cancer. Funct Integr Genomics. 2019;19(2):311-327. doi:10.1007/s10142-018-0641-9

Han N, Li Z. Non-coding RNA Identification in Osteonecrosis of the Femoral Head Using Competitive Endogenous RNA Network Analysis. Orthop Surg. 2021;13(3):1067-1076. doi:10.1111/os.12834

  1. Ln 270-375: Please indicate the references for the human and rodent studies that are congruent with the DEG/qRT-PCR expression patterns observed in the dataset.

Thank you for your comment on the omission  in lines 370-375 which has been already made up. Please find the references for the human and rodent studies that are correspond to the genes in our dataset below.

References:

Labonté B et al., Sex-specific transcriptional signatures in human depression. Nat Med. 2017 Sep;23(9):1102-1111. doi: 10.1038/nm.4386.

Cabrera-Mendoza B et al., Sex differences in brain gene expression among suicide completers. J Affect Disord. 2020 Apr 15;267:67-77. doi: 10.1016/j.jad.2020.01.167.

Han L, et al., ITGB4 deficiency in bronchial epithelial cells directs airway inflammation and bipolar disorder-related behavior. J Neuroinflammation. 2018 Aug 31;15(1):246. doi: 10.1186/s12974-018-1283-5.

Hlavacova N et al., Subchronic treatment with aldosterone induces depression-like behaviours and gene expression changes relevant to major depressive disorder. Int J Neuropsychopharmacol. 2012 Mar;15(2):247-65. doi: 10.1017/S1461145711000368.

Lee PH et al., Multi-locus genome-wide association analysis supports the role of glutamatergic synaptic transmission in the etiology of major depressive disorder. Transl Psychiatry. 2012 Nov 13;2(11):e184. doi: 10.1038/tp.2012.95.

Petrykey K et al., Influence of genetic factors on long-term treatment related neurocognitive complications, and on anxiety and depression in survivors of childhood acute lymphoblastic leukemia: The Petale study. PLoS One. 2019 Jun 10;14(6):e0217314. doi: 10.1371/journal.pone.0217314.

Klempan TA et al., Altered expression of genes involved in ATP biosynthesis and GABAergic neurotransmission in the ventral prefrontal cortex of suicides with and without major depression. Mol Psychiatry. 2009 Feb;14(2):175-89. doi: 10.1038/sj.mp.4002110.

  1. While it is informative to read about the list of changes that the authors reviewed in the discussion, the reviewer feels that a short paragraph describing how they think these pathways could alter DMPFC network output leading to depression and/or suicide will tremendously help the readers. Without such a model, the discussion just reads like a list of changes. While such a model would be highly speculative, it will nevertheless create a more easily digestible message from the paper.

To present a speculative model how the detected changes might actually contribute to suicidal behavior, we added the following sentences to the Conclusion: “Due to the alteration in pathways relating to gliogenesis, suicide behavior can affect glial cell numbers and cellular morphology in the medial part of the prefrontal cortex. Additionally, alterations in glutamate neurotransmission including both ionotropic and metabotropic receptors as well as neuroinflammatory processes potentially contributing to neuroplastic changes can contribute to the dysfunction of the DMPFC. The DMPFC is one of the nodes in connections with different cortical networks receiving direct viscero- and somatosensory inputs, motivational/emotional inputs from the orbitofrontal cortex, and cognitive inputs from the dorsolateral and ventral prefrontal cortex. By processing all this information together by activated working memories, the DMPFC may work out motor programs and projects (through the default mode network) to the precuneus and directly to the premotor and supplementary motor cortical areas. The precuneus is able to interfere or even block different actions of the motor cortex, and through self-reflection may determine suicidal behavior.”

  1. Ln 751-755: Considering the heterogeneity of cell types that the samples were collected from, authors should not make claims about causation. While the reviewer agrees that the authors’ data has several commonalities with other studies looking at expression changes in MDD and suicide from other brain areas, this could just be a common secondary pathology arising from dysregulated network activity.

Based on the advice of the reviewer, the claim about causation was canceled from the text of the Conclusion. Instead, a caution about secondary pathology was included. The part of the manuscript reads in its revised form as follows: “The identified gene expression alterations may reveal dysregulated pathways and functional cascades involved in the pathophysiology of suicidal behavior. Therefore, we conclude that the DEGs and pathways identified in this study could be linked to depression albeit a common secondary pathology arising from dysregulated network activity is also possible. These results are supported by the large overlap of DEGs between previously reported genes altered in depressive disorders.”

  1. Supp Table 1: Considering the wide variety of pathologies (including brain pathologies) that the “control subjects” displayed compared to the suicide victims, can the authors discuss the effects these pathological changes may have had on gene expression. Some of the pathologies listed for the controls are chronic and would affect function and gene expression across brain changes. On the other hand, the suicide victim tissue is largely free of any pathological changes.

We agree with the reviewer that chronic pathologies of control subjects may contribute to altered gene expression in brain tissue. Therefore, the same condition was not present in more than 50% of control subjects. Still, stroke may be the most serious issue in this regard. Therefore, we performed additional analysis to compare stroke and non-stroke patients.  Using a t-test, we found no significant difference while examining the gene expression level of control subjects with a history of stroke (n=4) and non-stroke subjects (n=4) for any of the genes selected for validation. Additionally, we also performed linear regression analysis for age- and PMI-dependent gene expression for all the validated genes, and we observed no significant correlation between the control individual’s age/PMI and the level of gene expression for 13 out of 14 genes.

MINOR POINTS:

  1. Ln 334-335: The sentence is incomplete.

Thank you for pointing out this mistake, which was corrected.

Reviewer 2 Report

My suggestions:

1. In the introduction, I would introduce briefly some genetic factors, which may impact the risk for psychiatric diseases.

2. For ClueGo figures (Figure 5), it would be better to upload in better resolution. Some parts are a little difficult to read.  In some parts, I suggest using different colors. 

3. In the methods, chapter "Human brain tissue samples", I would add a table, of how many controls and patients were used, and what were their agse. I would add on patients, what mental disease they were diagnosed with, and what age they were diagnosed with depression (if available). 

4. In discussion, I would add a table on over-and underexpressed genes, which have a common function, and in which depression-related pathways they act together.

5. Were any of these genes related to other neurodegenerative diseases, such as Alzheimer's disease or frontotemporal dementia?

Author Response

REVIEWER 2.

The critiques and suggestions of the Reviewers are highly appreciated. Based on these, the manuscript was substantially altered and improved. Please, find our point-by-point responses below:

  1. In the introduction, I would introduce briefly some genetic factors, which may impact the risk for psychiatric diseases.

Agreeing with the Reviewer that a brief summary of genetic factors contributing to the risk of depression is helpful, we added the following sentences to the Introduction (lines 78-86): “Genetic variants of serotonergic, dopaminergic and adrenergic genes are the most extensively studied genes in relation to both depression and suicidal behavior, however, to date there are only few candidate gene variants reliably associated with suicidality [43]. A recent meta-analysis identified novel depression-associated variants involved in the development and maintenance of cognitive processes [44], moreover a systematic review revealed that the most crucial candidate genes for MDD are involved in glutamate neurotransmission, regulation of calcium channel activity and apoptosis [45]. A cell type-specific methylome study pointed to the role of the innate immune responses via p75NTR/neurotrophic growth factor and toll-like receptor signaling in major depressive disorder (MDD) [46].” In addition, these parts of Introduction are referred to as corresponding parts of the revised Discussion (lines 562-564).

  1. For ClueGo figures (Figure 5), it would be better to upload in better resolution. Some parts are a little difficult to read.  In some parts, I suggest using different colors. 

Thank you for pointing out this shortcoming of the manuscript. The resolution of Figure 5 was improved to 2400 DPI and also the colors were changed to emphasize the major messages.

  1. In the methods, chapter "Human brain tissue samples", I would add a table, of how many controls and patients were used, and what were their ages. I would add on patients, what mental disease they were diagnosed with, and what age they were diagnosed with depression (if available). 

Description of the patients including all their major known parameters, such as sex, age, post-mortem interval, cause of death and clinical and pathological diagnosis was added to the main manuscript as Table 3.

  1. In discussion, I would add a table on over-and underexpressed genes, which have a common function, and in which depression-related pathways they act together.

We truly appreciate this comment of the Reviewer because it allowed us to create such a table, which we indeed find particularly informative. Accordingly, a new table, Table 2 was added to the Discussion. It contains down- and upregulated DEGs, which shared common functions with depression-related pathways, such as neuroactive ligand-receptor interaction, oxytocin signaling, GABAergic and Serotonergic synapse, and MAPK signaling pathway. In addition, the text of the Discussion was also supplemented with the following sentences: “Functional annotation of down- and upregulated DEGs revealed that there are gene sets overlapping common pathways associated with depression. Searching for genes, which share common functions with depression-related pathways, we found that both down- and upregulated genes are present among those, which participate in neurotransmission and cell metabolism processes taking place in the DMPFC.”

  1. Were any of these genes related to other neurodegenerative diseases, such as Alzheimer's disease or frontotemporal dementia?

To define the disease-associated differentially expressed genes we used the Gene-Disease Associations Dataset (GAD) approach. As a result of this analysis, we found over 100 neurodegenerative disease-associated genes related to Alzheimer’s and Parkinson’s disease, like SERPINA3, MT-ND4, MAOA or NOS3 among the DEGs we identified in the study.

Round 2

Reviewer 1 Report

All the reviewer's concerns have been addressed in this version of the manuscript.

Reviewer 2 Report

The authors fulfilled my suggestions, thank you